# Modeling Sleep Quality Depending on Objective Actigraphic Indicators Based on Machine Learning Methods

**DOI:** 10.3390/ijerph19169890

**Published:** 2022-08-11

**Authors:** Olga Vl. Bitkina, Jaehyun Park, Jungyoon Kim

**Affiliations:** 1Department of Industrial and Management Engineering, Incheon National University (INU), Academy-ro 119, Incheon 22012, Korea; 2Department of Computer Science, Kent State University, Kent, OH 44240, USA

**Keywords:** actigraphy, sleep quality, machine learning, support vector machine, k-nearest neighbors, naïve Bayes

## Abstract

According to data from the World Health Organization and medical research centers, the frequency and severity of various sleep disorders, including insomnia, are increasing steadily. This dynamic is associated with increased daily stress, anxiety, and depressive disorders. Poor sleep quality affects people’s productivity and activity and their perception of quality of life in general. Therefore, predicting and classifying sleep quality is vital to improving the quality and duration of human life. This study offers a model for assessing sleep quality based on the indications of an actigraph, which was used by 22 participants in the experiment for 24 h. Objective indicators of the actigraph include the amount of time spent in bed, sleep duration, number of awakenings, and duration of awakenings. The resulting classification model was evaluated using several machine learning methods and showed a satisfactory accuracy of approximately 80–86%. The results of this study can be used to treat sleep disorders, develop and design new systems to assess and track sleep quality, and improve existing electronic devices and sensors.

## 1. Introduction

Sleep disorders, including insomnia, are a group of disorders of the nervous system and manifest as problems of varying severity with falling asleep and maintaining sleep [1]. The consequences of sleep disorders manifest themselves negatively in various areas of life and affect its quality, cognitive function decreases, and impaired concentration of attention and memory. In the long term, this can lead to the development of anxiety, depression, and other physiological disturbances. Sleep problems are common in all populations, especially older people. Causes of sleep disorders include stress, anxiety, poor sleep hygiene, and overexcitation during the day. In general, these factors and human activity affect the duration of sleep, the rate of falling asleep, and the number and duration of awakenings [2]. Based on the relationship between human activity and objective sleep performance, this study proposes a classification model for sleep quality in relation to human behavioral performance during sleep.

To extract objective sleep metrics (e.g., duration of sleep, rate of falling asleep, and number and duration of awakenings), wrist actigraphy was used. Previous studies [3,4,5,6] reported actigraphy results, which were successfully used to build different models based on machine learning and other methods. A study [3] used actigraphy data from 39 cell lung cancer patients and found approximately 90% correspondence between actigraphy and sleep diaries of the patients. A previous study [6] presented actigraphy and video somnography methods for autism and nonspecific developmental delay research. Actigraphy was found to show 94% correspondence, 97% sensitivity, and 24% specificity compared to the somnography method. Based on the evidence obtained from the actigraphy method, this approach was used to model sleep quality as a function of objective sleep performance through machine learning methods.

Previous studies in various fields [7,8,9,10,11] have proven the effectiveness of these approaches in the building of classification models and improving their performance. Machine learning methods have been applied in research [9] to evaluate cognitive processes in artificial intelligence AI. It was found that people’s well-being could be increased after applying machine learning methods to explainable AI for decision-making tasks. Data storage can be improved in the cloud and big data computer systems [10]. An adversarial machine learning method was proposed to predict the hard drive health level. Research guidelines have been proposed to assess the scope of model explanation approaches [11]. Convolutional network models were successfully applied to predict the learned models. The acceptance (adoption) of new technologies depends on the trust of the prospective user and the level of stress during the interaction with new products. Based on this, researchers and designers have devoted their work to user modeling stress and trust using machine learning approaches. Previous studies have used signals as the base for emotional recognition and reported the following results: word recall tasks and human emotion recognition have been linked to the classification of positive and negative mental states [7]. The applied machine learning method (SVM) showed a classification accuracy of approximately 75.65%. A previous study [12] built a model based on machine learning to classify cognitive task performance. The accuracy of the proposed models ranged from 75% to 95%, depending on physiological signals. Research [8] showed the application of different mental and physiological tests (Trier social stress test, Trier mental challenge test, and Stroop test) to evaluate human emotions based on speech characteristics. The maximum accuracy was approximately 70%. The Driver database and object analysis were used to identify features for stressful state prediction. The obtained model showed an accuracy of 78.94%. The SVM method showed an accuracy of approximately 89% in detecting human stress during the Stroop color-word test, arithmetic test, and talking about stressful experiences or events [13]. In a comparison of previous and present studies, physiological signals from different sensors showed the potential to recognize human emotions and stress levels. Models based on this approach and machine learning methods show an average accuracy of approximately 70–90%, with scope for future improvement and development.

## 2. Literature Background of Sleep Performance Metrics

In 1995, a data source [14] found that actigraphy effectively detects and analyzes sleep disorders, including insomnia and hypersomnia. Measurements, such as the amount of time spent in bed, sleep duration, number of awakenings, and duration of awakenings, can effectively predict sleep disorders. Moreover, actigraphy is one of the main measurement tools used in different medical and research fields (Figure 1).

Figure 1 shows the primary application area of behavior and mental disorders, including sleep disorders. Actigraphy has proven to be effective in nervous, pathological, respiratory, heart/blood, and cancer diseases. The main actigraphy measurements used as independent variables in the present study and the model developed are shown in Table 1.

**Table 1 ijerph-19-09890-t001:** Context of independent variables.

Variables (Factor)	Context	References
Total minutes in bed	Minutes spent in bed per night	[15,16]
Total sleep time (TST)	Length of sleep per night expressed in minutes	
Wake after sleep onset (WASO)	Time spent awake after falling asleep for the first time	
Number of awakenings	Number of awakenings during the night	
Average awakening length	Time in seconds spent awakening during the night	
Movement index	The number of minutes without movement is expressed as a percentage of the movement phase (i.e., the number of periods with arm movement).	
Fragmentation index	The number of minutes with movement is expressed as a percentage of the immobile phase (i.e., the number of the period without arm movement)	
Sleep fragmentation index	The ratio of the movement and fragmentation indices	

Table 1 shows the actigraphy measurements partially used in previous studies to diagnose, classify, and predict sleep and nervous diseases. A study [3] examined the quality of sleep in patients with lung cancer. One study showed significant differences between awakenings for the evaluation of sleep disturbance using the actigraphy method. Research [4] has studied the effectiveness of insomnia assessment using actigraph data. After comparing the two groups of participants with and without insomnia, it was found that insomnia could be predicted by sleep onset latency, total sleep time, wake after sleep onset, sleep efficiency, and the number of awakenings. A previous study [5] compared actigraphy and polysomnography methods to evaluate chronic primary insomnia. Actigraphy has high potential for the treatment of chronic primary insomnia and should be used together with additional methods. Total wake time and sleep onset latency were underestimated, but total sleep time and sleep efficiency were overestimated using the actigraphy method. Actigraphy with video somnography was compared to the accuracy of nighttime awakening detection in children with a mean age of 47 months [6]. Actigraphy showed agreement between sleep onset time, sleep onset latency, total sleep time, number of awakenings, and number of nocturnal awakenings for the nighttime awakening assessment. Actigraphy was used for the detection of sleep–wake patterns in participants aged 4–7 years [17]. Actigraphy and sleep diaries can be used for the analysis of sleep start, sleep end, and assumed sleep. A previous study [18] has analyzed the effectiveness of cognitive behavioral therapy in the treatment of insomnia among college students. It was found that students who received therapy showed great improvement in sleep efficiency, sleep onset latency, number of awakenings, time awakened after sleep onset, insomnia severity, and global sleep quality based on actigraphy data. Actigraphy effectively evaluates and compares in-person and unguided Internet-delivered cognitive behavioral therapy for insomnia treatment among military personnel [19]. Control conditions used for the assessment were sleep efficiency, sleep onset latency, number of awakenings, wake time after sleep onset, Insomnia Severity Index, and Dysfunctional Beliefs and Attitudes about Sleep Scale. Based on the results obtained, both in-person and Internet methods are effective for the treatment of insomnia. However, in-person therapy demonstrated higher effectiveness than the Internet version for military personnel; in turn, the difference was not found for civilians. A previous study [20] analyzed sleep quality using actigraphy, which has proven the ability of actigraphy to detect primary insomnia among children with attention focused on the objective assessment of sleep duration. A study [21] has evaluated therapeutic approaches to evaluate behavioral insomnia. After cognitive behavioral therapy, children demonstrated significant improvements in sleep latency, wake after sleep onset, and sleep efficiency. Total sleep time was not correlated with the studied therapeutic approach.

Based on the above studies, it was found that direct measurements of an actigraph are effective indicators of sleep quality and can be used to monitor insomnia. In contrast to previous publications, the present study demonstrated a unique set of predictors (e.g., duration of sleep, rate of falling asleep, and number and duration of awakenings), which were compared with the subjective assessment of sleep quality through self-reported questionnaires from the experiment participants. This study shows that objective and subjective measurements can be an effective basis for building machine learning models to analyze sleep quality.

## 3. Methods

### 3.1. Data Source

This study was based on the Multilevel Monitoring of Activity and Sleep in Healthy People (MMASH) dataset released under the Open Database License v1.0 that is publicly available on PhysioNet [15,16,22]. The MMASH provides psychophysiological data related to the usual daily activities of participants. This open dataset contains cardiovascular measurements, responses to psychological questionnaires, sleep quality assessment, and movement and activity data for 24 h. Based on the Current Procedural Terminology coding requirements of the American Medical Association [23], actiograph data recording is most effective in practice for a period between 72 h and up to 14 days continuously. In our study, data were recorded for seven days, which is in line with these medical recommendations. These data were obtained using the actigraphy method, and the metrics shown in Table 1 were applied in this study. Actigraphy allows us to obtain objective data on the sleep process and the period of physical activity measured over a long period of time (up to several weeks) using an accelerometer placed on the arm [24,25,26]. Actigraphy is one of the most objective methods for assessing the sleep period since other methods are based on subjective people’s opinions (interview and survey). The objectivity of actigraphy is also ensured by the fact that this method is used in real-life conditions in contrast to experimental environments [27]. All data were collected in collaboration with industry (BioBeats Health Science Company) and academia (University of Pisa) by experts from different research areas, including health, psychophysiology, and neurology.

### 3.2. Methods

Twenty-two healthy young adult males (22–40 years old) were recruited for the experiment. Their weight and height were 60–115 kg and 169–205 cm, respectively. Researchers were looking for a homogeneous sample of subjects to minimize the differences between participants’ lifestyles and provide more objective results. The experimental participants signed official agreements after checking the study protocol, information on data usage, and risks (based on the General Data Protection Regulation: Regulation EU 2016/679 of the European Parliament and Council, 27/04/2016). This study was approved by the Ethics Committee of the University of Pisa (#0077455/2018). The participants held two devices continuously for 24 h (a heart rate monitor and an actigraph for evaluation of sleep quality). In this study, only actigraph (ActiGraph wGT3X-BT) data were used. ActiGraph wGT3X-BT (ActiGraph LLC, Pensacola, FL, USA) is a triaxial accelerometer to record and provide the physical activity of the participants. This device has dimensions of 4.6 × 3.3 × 1.5 and a weight of 19 g. Recording signal frequency ranges between 30 and 100 Hz. The sleep data provided were processed using the Cole–Kripke algorithm [28]. Additionally, for sleep quality, the Pittsburgh Sleep Quality Index (PSQI) was collected for each participant to analyze their sleeping patterns [29]. Basically, PSQI was developed to classify sleep quality into two categories: “good” and “poor” [30]. The evaluation of “good” and “bad” sleep quality is based on the participants’ self-rating of seven sleeping characteristics (subjective sleep quality, sleep latency, habitual sleep efficiency, sleep duration, awakening, sleep medication consumption, and daytime functioning). The index was summarized using scores ranging from 0 to 21. A score lower than “6” corresponds to good sleep quality. Previous studies [31,32,33] showed the efficiency of binary classification models based on PSQI in medical, education, and occupation research. It was found that binary classification can be applied to evaluate sleeping quality among healthcare workers during the COVID-19 pandemic [31], also among older adults experienced in inpatient treatment [32], and undergraduate students with different levels of academic performance [33]. Data provided by the actigraph contain the time when the participant went to bed and got out of bed; the time of falling asleep and needed to fall asleep; total sleeping time and time in bed; the ratio between total sleep and total in-bed time; time spent awake after falling asleep for the first time; the number of awakenings; awakening in minutes; time without/with movements; and the ratio between movement and fragmentation indices.

### 3.3. Signal Processing Algorithm

The Cole–Kripke algorithm [28] was used to process actigraph data to assess sleep quality. The Cole–Kripke algorithm comprises three main stages. The first stage is the recorded signal resampling into 1 min epoch intervals. The second stage is the second resampling of the actigraph signal from the defined epoch using the normalizing constant given by the following equation:(1)$$D\;(n)=C\;\sum_{i=-2}^{4}W_{i}\;X(n-i)$$
where D (*n*) is the output value and W*_i_* is the coefficient of multiplication of the resampled signal epoch value and normalizing constant C. The third stage of the algorithm involves the rescoring procedure for the output values obtained in Equation (1). During this algorithm application, it was possible to classify overnight sleep/no-sleep participant states. As a result, D (*n*) > 1 is in the no-sleep state and D (*n*) < 1 is in the sleep state. This algorithm is based on the partition of recorded data in 2 s intervals for further division into longer intervals to optimize the sleep/wake sliding window. Data accounting was performed with a maximum value of 30 s of activity per minute (values without overlapping). This algorithm proved its ability to separate sleeping and non-sleeping states with an accuracy of approximately 88% during the monitoring period. The actual actigraphic data for the percentage of sleep and latency showed accuracies of 82% and 90%, respectively. The Cole–Kripke algorithm demonstrated its effectiveness in determining adult sleep and non-sleeping states for use in research and medicine.

## 4. Analysis

### 4.1. Classification Approach and Model

Different machine learning methods were applied to evaluate the ability of actigraph data and PSQI to classify good and bad sleep quality. The proposed model was based on previous literature showing the satisfactory ability of actigraphy and PSQI to evaluate sleep problems (Figure 2).

### 4.2. Machine Learning Methods

To test the hypothesis and optimize the model of the dependence of perceived sleep quality on objective indicators, several machine learning methods, such as support vector machine (SVM), logistic regression (LG), KNN, and NB, were tested. Machine learning methods (e.g., neural networks, support vector machines, k-nearest neighbors [KNN], and naïve Bayes [NB]) have proven reliable, and standard modeling approaches in various fields, including medicine. Previous studies have demonstrated high machine learning performance in disease classification and diagnosis. A previous study [34] applied deep neural networks to classify skin cancer. A study [2] used artificial neural network architecture to detect eczema disease. The final best result demonstrated that the system had a 68.37% average confidence level for skin disease recognition. Previous research has shown that machine learning is one of the most accurate methods for diagnosing and classifying various diseases. Additionally, the correlation was tested between the sleep quality variable and each independent variable (actigraph data) using Pearson correlation within a significance level of 0.05 to assess the contribution of each predictor to the classification model. Previous studies have provided a basis for the hypothesis that sleep quality can be classified based on actigraphy data and machine learning techniques.

## 5. Results

The present study showed that the model developed contains the PSQI score as a dependent variable. Actigraph data are independent variables and contain the time when the participant went to bed and got up from bed; the time of falling asleep and needed to fall asleep; total sleeping time and time in bed; the ratio between total sleep and total in-bed time; time spent awake after falling asleep for the first time; the number of awakenings; awakening in minutes; time without/with movements; and the ratio between movement and fragmentation indices. The performance of the obtained model was validated based on k-fold cross-validation using 3/5/8 folds for the partition of the analyzed dataset. The performance of the developed sleep quality model was evaluated using the SVM, LG, KNN, and NB methods using the 3/5/8-fold cross-validation technique (Table 2, Table 3 and Table 4).

Based on Table 2, Table 3 and Table 4, the highest accuracy was shown by the SVM and KNN methods, with a range of 81–86% for different cross-validation proportions. Particularly, 5-fold cross-validation with an SVM accuracy of 86% provided the best classification results. Model performance metrics of PPV, sensitivity, and specificity showed heterogeneity of the results with a range of 0–100%. However, despite this diversity, in SVM and KNN models with better accuracy, all performance metrics also showed satisfactory results between 75% and 100%. The Pearson correlation showed that two predictors of time in bed and awakening in minutes have significance with correlation values of 0.437 and 0.526, respectively. Levels of the correlation coefficient in the range 0.4–0.6 are considered a “moderate association” at a significant level [35]. Based on this, the developed sleep classification model shows satisfactory performance and can be applied to sleep quality analysis.

## 6. Discussion

### 6.1. Validity of Applied Dataset and Machine Learning Methods

The above studies show that actigraphy is a method that allows one to isolate a set of metrics to assess physical conditions, including the quality of sleep. For these studies, various data sources (from previously open sources and experiments), mathematical methods (machine learning methods, factor analysis, and interviews), and evaluation metrics of model performance (accuracy, sensitivity, F1-score, and receiver operating characteristic curve) were used (Table 5).

In studies [36,37,38], MMASH actigraph data were used, which were also applied in this study, thus demonstrating the efficiency of the MMASH dataset. One study [36] used MMASH data to classify and monitor future heart rates using data from a wearable device through different machine learning and NN methods. Another study [37] proposed a survey and comparison of different models, including the MMASH-based model, to classify the perceived levels of loneliness and social isolation using logistic regression, random forest, and SVM methods. Both studies [36,37], with developed models based on actigraph metrics, achieved an accuracy of over 90%. A previous study [38] used heart rate variability segments from the MMASH dataset to predict wake/sleep state using combined shapelets and K-means algorithm with a model accuracy of over 77%. A higher model accuracy of 86% obtained in our study (Table 3) falls within the values between the results [36,37,38] and confirms the performance of the developed model in our research. The accuracy difference is because various mathematical variables were used in the models. A previous study [39] developed models to classify nocturnal awakenings based on actigraph data. The authors used statistical data, entropy, Poincaré plot features, total sleep time, wake after sleep onset, sleep-wake ratio, sleep latency, and sleep efficiency obtained during the experiment, with cohabiting couples where one of the participants had insomnia disorder. Models with accuracies between 75% and 80% were developed using random forest and SVM machine learning approaches. A previous study [40] experimented with actigraph data recording (raw accelerometer data, awake time, and summary of movement) to assess sleep quality. The models based on logistic regression, multilayer perception, convolutional neural network, recurrent neural network, and long-short-term memory cell approaches show accuracies of approximately 66–93%. Another study [41] used another publicly available actigraph data source to evaluate nocturnal awakenings. The metrics of entropy, statistical set, Poincaré plot features, total sleep time, wake after sleep onset, sleep-wake ratio, sleep efficiency, and complex correlation measures were used by applying random forest and SVM machine learning methods and showed an accuracy of approximately 73–84%. A study [42] conducted an experiment with undergraduate student participants and recorded multimodal data from smartphones and wearable devices, including an actigraph sensor. The authors applied a recurrent neural network model with long-short-term memory cells to predict sleep/wake and sleep onset/offset states with an accuracy of over 90%. A study [43] developed classification models with an accuracy of over 75% based on actigraph features of total sleep time, wake after sleep onset, sleep efficiency, and the number of awakenings during sleep laboratory experiments. In this study, machine learning methods of logistic regression, random forest, adaptive boosting, and extreme gradient boosting were used.

Previous research [36,37,38,39,40,41,42,43] proves the relationship between various sensory data and sleep parameters. However, in contrast to this literature, the present study links not only objective sensory measurements and sleep characteristics but also subjective (personal) metrics of perceived sleep quality (PSQI survey results). These dependencies are important from the point of view of the health state and mental well-being of a person, since it is the perceived state of sleep quality that determines the personal feeling of well-being [44]. Despite the fact that the data of such physiological sensors as an actiograph and a cardiograph provide sufficiently accurate data for assessing the sleep process, they reflect the physical state of a person to a greater extent. In turn, questionnaires (PSQI) allow us to evaluate the psychological component of the result of sleep (satisfaction and dissatisfaction) [44]. Studies [45,46] showed the medical application of binary subjective sleep quality assessment based on PSQI. Research [45] compared medical data of 80 patients with insomnia and a control group of 45 healthy participants. Authors reported that PSQI showed high reliability and good validity for patients with primary insomnia [45]. This result proves the ability of binary subjective sleep quality assessment to be used for sleep disorder detection. Mental health state was discovered based on PSQI in a study [46] using linear and binary logistic regressions. The authors collected PSQI and geriatric depression scale scores of elderly people in nursing homes to find the relations between subjective sleep quality and depression symptoms. The study supported the hypothesis that poor sleep quality is associated with increased depression signs in the elderly. Based on this, the authors presented subjective sleep quality as one of the depression features. This is very important for identifying and diagnosing mental problems (anxiety, depression, and stress).

Summarizing the comparative results between the previous models and the one developed in this study, it can be seen that the actigraph dataset used (particularly the MMASH dataset), machine learning methods, and the final performance proof confirm the effectiveness of the proposed method.

### 6.2. Model Performance

Based on the results obtained, two models, SVM and KNN, show better performance among the assessed models. Several studies have discussed these methods and noted their high accuracy and ability to be used in prediction models in different sciences. A study [47] evaluated the performance of an SVM using a new kernel approach for data mining. The performance analysis consisted of three stages. First, data mining items were evaluated using an SVM based on various kernel functions (linear, polynomial, radial, and sigmoid types). Second, feature vector optimization is applied to obtain the best performance and accuracy of the model. Third, the optimal kernel approach with the highest accuracy was extracted and compared with the four existing approaches. The new method showed the best performance when applied to an SVM. It was found that SVM has good accuracy; however, the performance metrics of SVM can be improved by the application and combination of additional approaches, such as kernels and dynamically growing self-organizing trees. A study [48] applied SVM and artificial neural networks to the classification problem of patients with and without heart attack experience. The overall result showed high performance of the applied methods with an accuracy of over 80%; in turn, the accuracy of the SVM showed better classification ability. In a previous study [49], a linear SVM was applied for student performance classification. The SVM model showed high performance with race, gender, and access to lunch as predictors. A previous study [50] demonstrated the high performance of the KNN method for breast cancer classification. KNN was compared with the decision tree machine learning method and showed that KNN was more effective in disease classification applications.

It was shown that the model of binary classification of sleep quality has satisfactory performance and can be used for medical or scientific purposes. Perceived sleep quality can be assessed based on objective actigraph data of time when the participant went to bed and got up from bed; time of falling asleep and needed to fall asleep; total sleeping time and time in bed; the ratio between total sleep and total in-bed time; time spent awake after falling asleep for the first time; the number of awakenings; awakening in minutes; time without/with movements; and the ratio between movement and fragmentation indices. Previous studies have used different combinations of actigraph data and have demonstrated their modeling ability for classification tasks. One study [51] found that sleep quality is characterized by duration, rhythm, and quality. The sleep cycle was analyzed using actigraphy data and questionnaire indices. The analysis showed that the characteristics of regularity, fragmentation, active phase, relative, and rest amplitude of sleep could be used for sleep quality evaluation. Another study [52] analyzed the actigraphic ability to evaluate sleep quality. Actigraphic data was found to play an important role in health research for sleep-monitoring tasks. Actigraphy is a simple and cost-effective method for analyzing the sleep–wake process and evaluating sleep disorders, such as insomnia. Furthermore, the data provided can be used in the treatment and observation processes of sleep disorders. One study [53] reported sleep assessment results comparing the developed smartphone apps and objective actigraphy data. As a result, there were no significant differences between the methods, except for total sleep time measurements. The app overestimated the total sleep time; however, all approaches showed satisfactory performance.

The Pearson correlation showed the association between sleep quality and time in bed, as well as awakening minutes at a significant level. This result can be explained by the fact that these two predictors are directly related to the sleep process. Time spent in bed and awakening in minutes were identified in previous studies as symptoms associated with sleep disorders and may also be signs of insomnia [54]. Time in bed also can be considered a sign of poor sleep quality if the patient stays asleep less than 85% of this time [55]. Moreover, awakenings are associated with insomnia disorder, disturb the quality of sleep, and can be the reason for problems with sleeping [56]. Generally, all used model predictors showed potential for determining and solving sleep problems [4,5,6]. Some research [4] has shown that insomnia could be classified by natural patient data of sleep onset latency, total sleep time, wake after sleep onset, sleep efficiency, and the number of awakenings. The study in [5] showed efficiency in the model predictors of wake time and sleep onset in chronic primary insomnia evaluation. Nighttime awakening can be detected in children with a mean age up to 5 years old [6]. Predictors of sleep onset time, sleep onset latency, total sleep time, number of awakenings, and number of nocturnal awakenings showed this ability. Based on the combination of previous findings and presented results, it was found that the developed model contains effective predictors to evaluate and manage sleeping disorders, including insomnia.

The present study confirms and complements previous studies on sleep quality. Data acquisition and analysis methods confirmed the publications discussed above; however, the developed model contained unique independent variables obtained from the actigraph that could predict the subjective assessment of the sleep quality of participants or patients. The obtained model predictors included the time when the participant went to bed and got up from bed; time of falling asleep and needed to fall asleep; total sleeping time and time in bed; the ratio between total sleep and total in-bed time; time spent awake after falling asleep the first time; the number of awakenings; awakening in minutes; time without/with movements; and the ratio between movement and fragmentation indices. The results obtained can be practically used in medicine to improve the diagnosis and treatment of sleep disorders, develop and design sensors for medical purposes, and develop new electronic applications for computers and smartphones to monitor the state of human health.

### 6.3. Limitation, Application, and Future Research

Despite the satisfactory results of the accuracy of the developed model, the presented study has several limitations. The first is the number and grouping of participants in the experiment. The experiment involved 22 males; all participants were considered conditionally healthy without sleep disorders and mental problems. In order to universalize the study and build models that are more difficult to classify, it is planned to expand the number of participants and include female participants and participants with sleep disorders such as insomnia or sleep anxiety in the experiment. Second, the perceived quality of sleep was measured by the PSQI questionnaire that was designed to define only two conditions: good and poor [30]. This parameter allowed us to build a binary model of sleep quality based on machine learning methods. In the future, it is planned to use several sleep questionnaires to expand the classification groups in mathematical modeling. Third, the presented research applied one type of objective physiological data: actigraph measures. This approach can be extended to use additional sensors to collect various types of physiological responses, for example, heart rate and galvanic skin response. The application of different sensory objective measures will be useful to improve the developed model. Additionally, in future research, it is planned to apply additional methods of deep learning, for example, a convolutional neural network. This will improve the analytical approach and find additional dependencies between objective and subjective measurements.

The application of the obtained results and the developed model is possible in various theoretical and practical areas. Firstly, it expands knowledge in medical, behavioral, social, and physiological sciences, including sensors of human physiological signals (actigraph, cardiograph, and various stress indicators). New knowledge provides additional connections and dependencies between objective and subjective human data and assessments and also allows for a better understanding of perceived well-being and health. The main area of the practical application of the results obtained is monitoring and developing additional approaches to diagnose and track problems with the sleep process, as well as various mental, behavioral, and physiological disorders. This is possible because many health problems are accompanied by poor sleep quality, such as anxiety, depression, eating disorders, as well as cancer, and neurological diseases [44,57].

## 7. Conclusions

In this study, a cross-validated model is proposed to perceive the classification of sleep quality (satisfied and unsatisfied) based on objective actigraph data, including the time when the participant went to bed and got up from bed; time of falling asleep and needed to fall asleep; total sleeping time and time in bed; the ratio between total sleep and total in-bed time; time spent awake after falling asleep the first time; the number of awakenings; awakening in minutes; time without/with movements; and the ratio between movement and fragmentation indices. Two machine learning approaches (SVM and KNN) showed satisfactory performance metrics and accuracy. Objective actigraphy data can be an accurate predictor of sleep quality based on human activity. The following conclusions were drawn.

The developed model showed satisfactory classification ability and demonstrated the mutual connection between sleeping, human activity, and actigraph data.The proposed model applied to the real actigraph dataset showed satisfactory performance with an accuracy of approximately 80%. This result is consistent with previous studies using the same MMASH dataset.Machine learning methods (SVM and KNN) showed better performance than LR and NB.The combination of actigraph features can be used to access the human sleep process and predict sleep disorders.

The results obtained can be used for theoretical and practical applications. This study provides new knowledge for activity sensors, medicine, and behavioral/physiological sciences. The developed model and results will help adapt new schemes for predicting and treating sleep disorders, find basic connections between human activity and sleep quality, and expand the use of actigraphs in medicine and science. The insights from this study could serve medical professionals to improve the treatment process of sleep disorders and sensor developers to increase the performance and use areas of actigraphs. Designers can also introduce their findings to human/machine-interacting systems, wristbands, tablets, and laptops.

## Figures and Tables

**Figure 1 ijerph-19-09890-f001:**
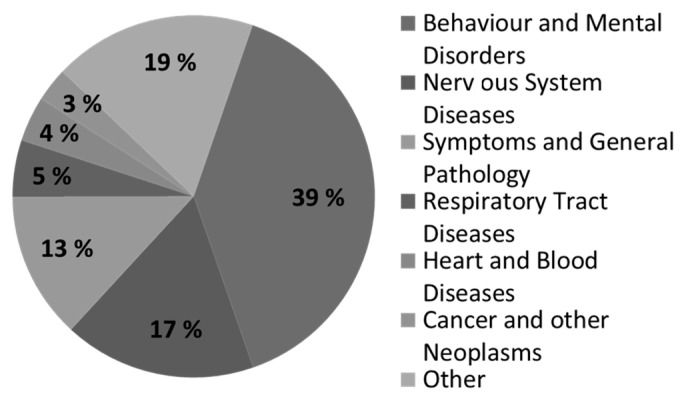
Actigraphy application area.

**Figure 2 ijerph-19-09890-f002:**
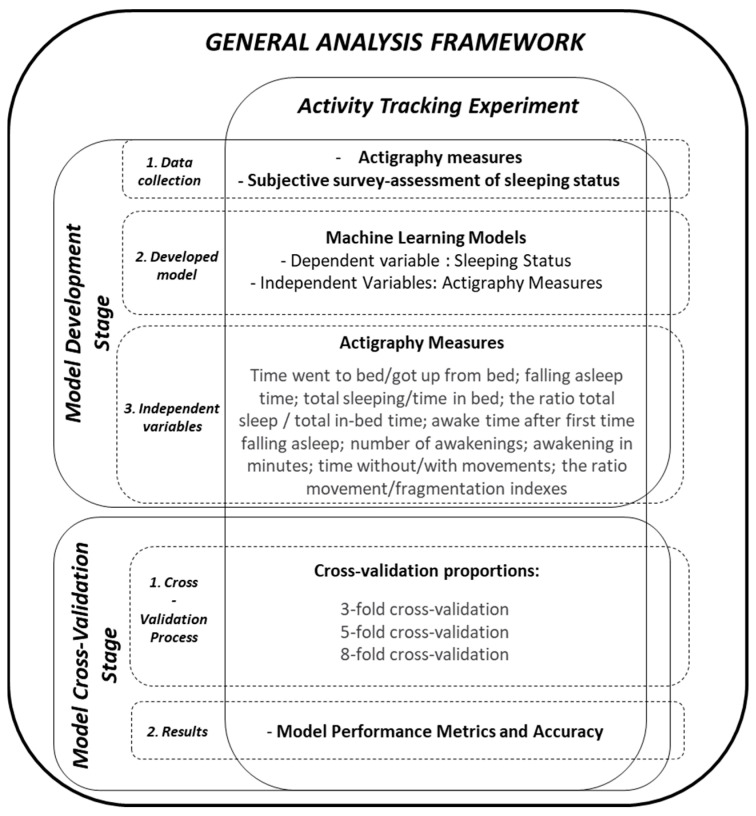
Model development process.

**Table 2 ijerph-19-09890-t002:** Classifiers’ comparison based on 3-fold cross-validation.

Classifier	Accuracy	PPV	Sensitivity	Specificity
Logistic regression	57%	60	75	33
Support vector machine	71%	100	71	0
Fine k-nearest neighbor	81%	100	79	100
Naïve Bayes	67%	93	70	0

**Table 3 ijerph-19-09890-t003:** Classifiers’ comparison based on 5-fold cross-validation.

Classifier	Accuracy	PPV	Sensitivity	Specificity
Logistic regression	62%	60	82	40
Support vector machine	86%	93	88	80
Fine k-nearest neighbor	76%	93	78	67
Naïve Bayes	67%	93	70	0

**Table 4 ijerph-19-09890-t004:** Classifiers’ comparison based on 8-fold cross-validation.

Classifier	Accuracy	PPV	Sensitivity	Specificity
Logistic regression	67%	80	75	40
Support vector machine	71%	100	71	0
Fine k-nearest neighbor	81%	93	82	75
Naïve Bayes	71%	100	71	0

**Table 5 ijerph-19-09890-t005:** Research comparison.

Study	Dataset Used	Machine Learning Methods	Independent Variables	Dependent Variables	Average Model Accuracy
[36]	Open source MMASH	Autoregressive integrated moving average, linear regression, support vector regression, K-nearest neighbor, decision tree, random forest, and long-short-term memory	Heart rate time-series	Expected heart rate	Over 90%
[37]	Cross-disciplinary survey using open source MMASH and other	Logistic regression, random forest, support vector machine	Different metrics of wireless technology and wearables	Perceived loneliness, social isolation levels	Over 90%
[38]	Open source MMASH	Combined shapelets and K-means algorithm	Heart rate variability segment	Wake/sleep state	Over 77%
[39]	Experiment with co-habiting couples	Random forest, support vector machine	Entropy, statistics, Poincaré plot features, total sleep time, wake after sleep onset, sleep-wake ratio, sleep latency and sleep efficiency	Nocturnal Awakenings	Approximately 75–80%
[40]	Experiment with random participants	Logistic regression, multilayer perception, convolutional neural network, recurrent neural network, a long-short-term memory cell	Raw accelerometer data, awake time, a summary of movements	Sleep quality	Approximately 66–93%
[41]	Publicly available source	Random forest, support vector machine	Entropy, statistics, Poincaré plot features, total sleep time, wake after sleep onset, sleep-wake ratio, sleep efficiency, and complex correlation measure	Nocturnal awakenings	Approximately 73–84%
[42]	Experiment with undergraduate students	Recurrent neural network with long-short-term memory cells	Different combinations of multimodal data from smartphones and wearable technologies	Sleep/wake state, sleep onset/offset	Over 90%
[43]	Experiment in a sleep laboratory	Logistic regression, random forest, adaptive boost, and extreme gradient boost	Total sleep time, wake after sleep onset, sleep efficiency, number of awakenings	Wake/sleep state	Over 75%

## Data Availability

The data used for this study are available upon request.

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
