# Peer review of "Modeling Sleep Quality Depending on Objective Actigraphic Indicators Based on Machine Learning Methods"

_ijerph, 2022, doi:10.3390/ijerph19169890_

Round 1

Reviewer 1 Report

The authors demonstrate that they can predict perceived sleep quality using actigraphy data with nearly 80% accuracy. The study looks fine and and the methods used are appropriate. Though, I have questions about the novelty and contributions.

1) What is the gap being addressed by this study. Looks like there is already a great deal of work (as outlined in Table 5) that predict sleep quality using similar measures. I am not sure what the authors are trying to address. I haven't seen anywhere in the paper that this study addresses a gap in the literature.

2) Why is it important to predict perceived sleep quality? Can the authors provide a use case (or an example) where this could be important? The reason I ask is: the actigraphy device already provides information about sleep quality. One can look at the data and make judgments about a person's sleep quality. In fact, many studies use the data collected by actigraphy devices as a dependent variable (because it can generate data that can be used for "sleep quality"). Instead, the authors measured perceived sleep quality and tried to predict that instead. Further, perceived sleep quality is a binary variable, which is a yes/no variable. So, if the authors can make a case as to why one might want to know perceived sleep quality (as a yes/no variable), then they can make a stronger case for the paper. Currently, the paper does not mention anything about why this predictive model is important and how it can be used in a clinical (or in another) setting.

Author Response

Comments and Suggestions for Authors

The authors demonstrate that they can predict perceived sleep quality using actigraphy data with nearly 80% accuracy. The study looks fine and the methods used are appropriate. Though, I have questions about the novelty and contributions.

  • What is the gap being addressed by this study. Looks like there is already a great deal of work (as outlined in Table 5) that predict sleep quality using similar measures. I am not sure what the authors are trying to address. I haven't seen anywhere in the paper that this study addresses a gap in the literature.

Answer) The authors appreciate the reviewer for the kind opinion and all comments. The authors agree that in previous studies (eg, Table 5, as noted by the reviewer), the issue of sleep quality has also been raised and associated with objective measures of activity. The main distinguishing feature of the presented study from the previous ones is the hypothesis of the relationship between objective (sensor data) and subjective indicators of a person (survey) for assessing the quality of sleep. This is important for several reasons; firstly, the subjective assessment of sleep quality is very important since it reflects the real feeling of well-being of a person since this is an individual perceived metric. Therefore, the subjective sleep assessment reflects the perceived feeling of a human health state. In turn, it is not always possible to obtain both subjective and objective indicators at the same time in real practice, but we need to know how each of these metric groups is related to each other in order to be able to predict this relationship if there is a lack of data on the individual opinion and well-being feelings of a person. Based on this, it is necessary to build models that combine subjective and objective methods to develop more accurate and comprehensive analysis approaches. In turn, in previous studies, it is difficult to find an assessment and analysis of this kind of approach, and our article is intended to cover this omission.

     According to reviewer’s comment, we added an explanation as follows (please see page 11 and lines 25-36): “Previous research [36-43] proves the relationship between various sensory data and sleep parameters. However, in contrast to this literature, the present study links not only objective sensory measurements and sleep characteristics, but also subjective (personal) metrics of perceived sleep quality (PSQI survey results). These dependencies are important from the point of view of the health state and mental well-being of a person, since it is the perceived state of the sleep quality that determines the personal feeling of well-being [44]. Despite the fact that the data of such physiological sensors as an actiograph and a cardiograph provide sufficiently accurate data for assessing the sleep process, however, they reflect the physical state of a person to a greater extent. In turn, questionnaires (PSQI) allow us to evaluate the psychological component of the result of sleep (satisfaction and dissatisfaction) [44]. This is very important for identifying and diagnosing mental problems (anxiety, depression, and stress)”.

  • Why is it important to predict perceived sleep quality? Can the authors provide a use case (or an example) where this could be important? The reason I ask is: the actigraphy device already provides information about sleep quality. One can look at the data and make judgments about a person's sleep quality. In fact, many studies use the data collected by actigraphy devices as a dependent variable (because it can generate data that can be used for "sleep quality"). Instead, the authors measured perceived sleep quality and tried to predict that instead. Further, perceived sleep quality is a binary variable, which is a yes/no variable. So, if the authors can make a case as to why one might want to know perceived sleep quality (as a yes/no variable), then they can make a stronger case for the paper. Currently, the paper does not mention anything about why this predictive model is important and how it can be used in a clinical (or in another) setting.

Answer) Thanks to the reviewer for the comment. The answer to this question is somewhat related to our previous answer to the first comment. The perceived quality of sleep is one of the important health characteristics used in psychiatry and the diagnosis of serious physical (not only mental) diseases (e.g. cancer). The authors agree that the actiograph is an important source of data, but it can assess the objective physical condition of a person. In turn, perceived values, including the quality of sleep, play an important role in assessing the individual state of the patient's psyche. As some examples (according to reviewer’s comment), this measure was used in clinical studies such as Huang Y, Zhu M. “Increased Global PSQI Score Is Associated with Depressive Symptoms in an Adult Population from the United States”, 2020; and Momayyezi M, Fallahzadeh H, Farzaneh F, Momayyezi M. “Sleep Quality and Cancer-Related Fatigue in Patients with Cancer”, 2021. The authors confirm that the perceived quality of sleep is important in assessing mental and physical states. Based on this, the combination of subjective and objective methods of assessing the quality of sleep helps to determine with greater accuracy the physical and mental states at the same time and contributes to the diagnosis of health (including insomnia, depression, mood disorders, cancer, etc.).

     According to the reviewer’s comment, we added an explanation on the page 11 and lines 25-36 (please also see citation of added and revised text in our above answer).

    Additionally, to improve part of the analysis in our study and proof the obtained result's importance for perceived sleep quality, we performed an additional Pearson correlation analysis between perceived sleep quality and actigraph predictors and added results with explanations. Pearson correlation analysis was introduced as follows (please see page 8 and lines 13-16): “Additionally, the correlation was tested between the sleep quality variable and each independent variable (actigraph data) using Pearson correlation within a significance level of 0.05 to assess the contribution of each predictor to classification model”.

     Results of this new Pearson analysis showed the following (please see the page 9 and lines 35-38): “The Pearson correlation showed that two predictors of time in bed and awakening in minutes have significance with correlation values of 0.437 and 0.526 respectively. Levels of the correlation coefficient between 0.4 – 0.6 are considered as a “moderate association” at a significant level [35]”.

     Also, the authors showed more perceived sleep quality model reliability based on Pearson analysis in sub-section “6.2. Model Performance” as follows (please see page 12 and lines 36-53): “The Pearson correlation showed the association between sleep quality and time in bed, as well as awakening minutes at a significant level. This result can be explained by the fact that these two predictors are directly related to the sleep process. Time spent in bed and awakening in minutes were identified in previous studies as symptoms associated with sleep disorders and may also be signs of insomnia [52]. Time in bed also can be considered a sign of poor sleep quality if the patient stays asleep less than 85% of this time [53]. Moreover, awakenings are associated with insomnia disorder, disturb the quality of sleep, and can be the reason for problems with sleeping [54]. Generally, all used model predictors showed potential for determination and solving the sleep problems [4-6]. Research [4] showed that insomnia could be classified by natural patient data of sleep onset latency, total sleep time, wake after sleep onset, sleep efficiency, and the number of awakenings. Research [5] showed efficiency in the model predictors of wake time and sleep onset in chronic primary insomnia evaluation. Nighttime awakening can be detected in children with a mean age up to 5 years old [6]. Predictors of sleep onset time, sleep onset latency, total sleep time, number of awakenings, and number of nocturnal awakenings showed this ability. Based on the combination of previous findings and presented results, it was found that the developed model contains effective predictors to evaluate and manage the sleeping disorders including insomnia”.

Reviewer 2 Report

1. In the future clinical practice of the author's model, what time span of actigraphy data is enough to access the sleep quality of patients?

2. It isn't recommended to cite too many literatures in the methods section. The authors should  introduce the background and quotation in the introduction section.

3. The authors divided sleep quality into two categories: good and bad. It is suggested that the classification should be more detailed。

4. Some spelling error should be corrected, and English language should be improved.

Author Response

Comments and Suggestions for Authors

  1. In the future clinical practice of the author's model, what time span of actigraphy data is enough to access the sleep quality of patients?

Answer) The authors appreciate the reviewer for the kind opinion and all comments. Basically, the recommended duration of actigraphy recording is a minimum of 72 hours to 14 consecutive days, in accordance with the Current Procedural Terminology (CPT) coding requirements of the American Medical Association. In turn, our research is based on wearable activity tracker data collected during 7 consecutive days. Based on this, the proposed model is consistent with the requirements of the American Medical Association. This information was added as follows (Please see Page 5; Lines 8-12): “Based on the Current Procedural Terminology coding requirements of American Medical Association [23], actiograph data recording is most effective in practice for a period between 72 hours and up to 14 days continuously. In our study, data were recorded for 7 days, which is in line with these medical recommendations”.

  1. It isn't recommended to cite too many literatures in the methods section. The authors should introduce the background and quotation in the introduction section.

Answer) Thanks to the reviewer for the comment. According to reviewer’s comment, the authors revised and reduced the Method section and put the majority of cited literature in the Introduction section as follows ( Please see page 2 Lines 3-32; and Page 8, Lines 2-17):” Previous studies in various fields [7–11] have proven the effectiveness of these approaches in the building of classification models and improving their performance. Machine learning methods have been applied in research [9] to evaluate cognitive processes in artificial intelligence AI. It was found that people’s well-being could be increased after applying machine learning methods to explainable AI for decision-making tasks. Data storage can be improved in the cloud and big data computer systems [10]. An adversarial machine learning method was proposed to predict the hard drive health level. Research guidelines have been proposed to assess the scope of model explanation approaches [11]. Convolutional network models were successfully applied to predict the learned models. The acceptance (adoption) of new technologies depends on the trust of the prospective user and the level of stress during the interaction with new products. Based on this, researchers and designers have devoted their work to user modeling stress and trust using machine learning approaches. Previous studies have used signals as the base for emotional recognition and reported the following results: word recall tasks and human emotion recognition have been linked to the classification of positive and negative mental states [7]. The applied machine learning method (SVM) showed a classification accuracy of approximately 75.65%. A previous study [12] built a model based on machine learning to classify cognitive task performance. The accuracy of the proposed models ranged from 75% to 95%, depending on physiological signals. Research [8] showed the application of different mental and physiological tests (Trier social stress test, Trier mental challenge test, and Stroop test) to evaluate human emotions based on speech characteristics. The maximum accuracy was approximately 70%. Driver database and object analysis to identify features for stressful state prediction. The obtained model showed an accuracy of 78.94%. The SVM method showed an accuracy of approximately 89% in detecting human stress during the Stroop color-word test, arithmetic test, and talking about stressful experiences or events [13]. In a comparison of previous and present studies, physiological signals from different sensors showed the potential to recognize human emotions and stress levels. Models based on this approach and machine learning methods show an average accuracy of approximately 70-90%, with scope for future improvement and development.”;

     “To test the hypothesis and optimize the model of the dependence of perceived sleep quality on objective indicators, several machine learning methods, such as support vector machine (SVM), logistic regression (LG), KNN, and NB, were tested. Machine learning methods (e.g., neural networks, support vector machines, k-nearest neighbors [KNN], and naïve Bayes [NB]) have proven reliable and standard modeling approaches in various fields, including medicine. Previous studies have demonstrated high machine learning performance in disease classification and diagnosis. A previous study [34] applied deep neural networks to classify skin cancer. A study [2] used artificial neural network architecture to detect eczema disease. The final best result demonstrated that the system had a 68.37% average confidence level for skin disease recognition. Previous research has shown that machine learning is one of the most accurate methods for diagnosing and classifying various diseases. Additionally, the correlation was tested between the sleep quality variable and each independent variable (actigraph data) using Pearson correlation within a significance level of 0.05 to assess the contribution of each predictor to the classification model. Previous studies have provided a basis for the hypothesis that sleep quality can be classified based on actigraphy data and machine learning techniques”.

  1. The authors divided sleep quality into two categories: good and bad. It is suggested that the classification should be more detailed。

Answer) Thanks to the reviewer for the suggestion. The authors agree that increasing the classification categories will improve the developed model, but the data used impose several limitations in the current study. Firstly, our depending variable is based on Pittsburgh quality sleep Index. This index was originally developed to classify sleep quality only into two categories: good and poor. Criteria are - if a score is higher than 5 - it's poor sleep. Physionet experimental data source demonstrates: "PSQI: Pittsburgh Sleep Quality Questionnaire Index gives a score rating from 0 to 21, with values lower than 6 indicating good sleep quality". Based on this, basically our dependent variable allows classifying data using a binary approach. The second reason is the number of participants. More classes will cause separation of 22 participants into smaller groups, which is difficult to balance due to the data limitation. According to reviewer’s comment, we added an explanation of this research limitation in the new sub-section “6.3. Limitation, Application, and Future Research” (please see page 13 and lines 14-44) to discuss our results with future improvement as follows: “Despite the satisfactory results of the accuracy of the developed model, the presented study has several limitations. The first is the number and grouping of participants in the experiment. The experiment involved 22 male; all participants were considered conditionally healthy without sleep disorders and mental problems. In order to universalize the study and build models that are more difficult to classify, it is planned to expand the number of participants, and include female participants and participants with sleep disorders such as insomnia or sleep anxiety in the experiment. Second, perceived quality of sleep was measured by PSQI questionnaire that was designed to define only two conditions - good and poor [55]. This parameter allowed us to build a binary model of sleep quality based on machine learning methods. In the future, it is planned to use several sleep questionnaires to expand the classification groups in mathematical modeling. Third, the presented research applied one type of objective physiological data – actigraph measures. This approach can be extended to use additional sensors to collect various types of physiological responses, for example, heart rate and galvanic skin response. The application of different sensory objective measures will be useful to improve the developed model. Also, in future research, it is planned to apply additional methods of deep learning, for example, a convolutional neural network. This will improve the analytical approach and find additional dependencies between objective and subjective measurements. The application of the obtained results and the developed model is possible in various theoretical and practical areas. Firstly, it expands knowledge in medical, behavioral, social and physiological sciences, including sensors of human physiological signals (actigraph, cardiograph and various stress indicators). New knowledge provides additional connections and dependencies between objective and subjective human data and assessments, and also allows for a better understanding of perceived well-being and health. The main area of the practical application of the results obtained is monitoring and developing additional approaches to diagnosing and tracking problems with the sleep process, as well as various mental, behavioral and physiological disorders. This is possible because many health problems are accompanied by poor sleep quality, such as anxiety, depression, eating disorders, as well as cancer and neurological diseases [44, 56]”.

    However, to improve part of the analysis in our study, we performed an additional Pearson correlation analysis and added results with explanations. Pearson correlation analysis was introduced as follows (please see page 8 and lines 13-16): “Additionally, the correlation was tested between the sleep quality variable and each independent variable (actigraph data) using Pearson correlation within a significance level of 0.05 to assess the contribution of each predictor to classification model”. Results of this new Pearson analysis sowed the follows (please see the page 9 and lines 35-38): “The Pearson correlation showed that two predictors of time in bed and awakening in minutes have significance with correlation values of 0.437 and 0.526 respectively. Levels of the correlation coefficient between 0.4 – 0.6 are considered as a “moderate association” at a significant level [35]”. Also, the authors showed more model reliability based on Pearson analysis in sub-section “6.2. Model Performance” as follows (please see page 12 and lines 36-53): “The Pearson correlation showed the association between sleep quality and time in bed, as well as awakening minutes at a significant level. This result can be explained by the fact that these two predictors are directly related to the sleep process. Time spent in bed and awakening in minutes were identified in previous studies as symptoms associated with sleep disorders and may also be signs of insomnia [52]. Time in bed also can be considered a sign of poor sleep quality if the patient stays asleep less than 85% of this time [53]. Moreover, awakenings are associated with insomnia disorder, disturb the quality of sleep, and can be the reason for problems with sleeping [54]. Generally, all used model predictors showed potential for determination and solving the sleep problems [4-6]. Research [4] showed that insomnia could be classified by natural patient data of sleep onset latency, total sleep time, wake after sleep onset, sleep efficiency, and the number of awakenings. Research [5] showed efficiency in the model predictors of wake time and sleep onset in chronic primary insomnia evaluation. Nighttime awakening can be detected in children with a mean age up to 5 years old [6]. Predictors of sleep onset time, sleep onset latency, total sleep time, number of awakenings, and number of nocturnal awakenings showed this ability. Based on the combination of previous findings and presented results, it was found that the developed model contains effective predictors to evaluate and manage the sleeping disorders including insomnia”.

     Also, we added an explanation about the reason for binary classification in the model with additional literature background as follows (Please see the page5, lines 40-50): “Basically PSQI was developed to classify sleep quality into two categories – “good” and “poor” [30]. The evaluation of “good” and “bad” sleep quality is based on the participants’ self-rating of seven sleeping characteristics (subjective sleep quality, sleep latency, habitual sleep efficiency, sleep duration, awakening, sleep medication consumption, and daytime functioning). The index was summarized using scores ranging from 0 to 21. A score lower than “6” corresponds to good sleep quality. Previous studies [31-33] showed the efficiency of binary classification models based on PSQI in medical, education, and occupation research. It was found that binary classification can be applied to evaluate sleeping quality among healthcare workers during the COVID-19 pandemic [31], also among older adults experienced in inpatient treatment [32], and undergraduate students with different levels of academic performance [33].”

  1. Some spelling error should be corrected, and English language should be improved.

Answer) The authors appreciate the reviewer for comment. This manuscript previously passed the English proofreading service and was checked.

Reviewer 3 Report

This study presents a methodology for modeling sleep quality using machine learning algorithms. The topic is interesting, however, some issues still need to be considered: 

Major concerns:

1.      Except for the comparison of model performance for the machine learning algorithms, more attention in the results and discussion sections needs to be paid on the model interpretability, contribution of predictors, and the potential biological mechanism, in order to show the reliability of your model.

2.      An external validation of your models in another population is suggested.

3.      Sleep quality is also affected by the activities and life style in the daytime, these predictors are suggested to be included in the model.

4.      More of the article should be devoted to analyzing the strengths and weaknesses of the model used for the sleep quality analysis in this paper.

 Minor concerns:

1.      The authors mentioned that “a homogeneous sample of subjects to minimize the differences between participants’ lifestyles and provide more objective results”, selection criteria of these participants and their characteristics should be presented.

2.      What is the population structure of the data source, and whether the model is suitable for all the population or only part of the population?

3.      How does the actigraph determine and classify sleep and awaken status, whether there are errors, and whether errors have been well adjusted?

4.      What is the limitation of the chose model, whether it can be further improved?

Author Response

Comments and Suggestions for Authors

This study presents a methodology for modeling sleep quality using machine learning algorithms. The topic is interesting. However, some issues still need to be considered: 

Major concerns:

  1. Except for the comparison of model performance for the machine learning algorithms, more attention in the results and discussion sections needs to be paid on the model interpretability, contribution of predictors, and the potential biological mechanism, in order to show the reliability of your model.

Answer) Thanks to the reviewer for the kind opinion and all the comments. According to the reviewer’s comment, to show more results on model interpretability, the contribution of predictors, and potential biological mechanism, the authors performed additional analysis as follows (please see page 8 and lines 13-16): “Additionally, the correlation was tested between the sleep quality variable and each independent variable (actigraph data) using Pearson correlation within a significance level of 0.05 to assess the contribution of each predictor to classification model”.

     Results of this new analysis sowed the following (please see the page 9 and lines 35-38): “The Pearson correlation showed that two predictors of time in bed and awakening in minutes have significance with correlation values of 0.437 and 0.526 respectively. Levels of the correlation coefficient between 0.4 – 0.6 are considered as a “moderate association” at a significant level [35]”.

     Authors showed more model reliability and discussed this by adding new text in sub-section “6.2. Model Performance” as follows (please see page 12 and lines 36-53): “The Pearson correlation showed the association between sleep quality and time in bed, as well as awakening minutes at a significant level. This result can be explained by the fact that these two predictors are directly related to the sleep process. Time spent in bed and awakening in minutes were identified in previous studies as symptoms associated with sleep disorders and may also be signs of insomnia [52]. Time in bed also can be considered as a sign of poor sleep quality if the patient stays asleep less than 85% of this time [53]. Moreover, awakenings are associated with insomnia disorder, disturb the quality of sleep, and can be the reason for problems with sleeping [54]. Generally, all used model predictors showed potential for determination and solving the sleep problems [4-6]. Research [4] showed that insomnia could be classified by natural patient data of sleep onset latency, total sleep time, wake after sleep onset, sleep efficiency, and the number of awakenings. Research [5] showed efficiency in the model predictors of wake time and sleep onset in chronic primary insomnia evaluation. Nighttime awakening can be detected in children with a mean age up to 5 years old [6]. Predictors of sleep onset time, sleep onset latency, total sleep time, number of awakenings, and number of nocturnal awakenings showed this ability. Based on the combination of previous findings and presented results, it was found that the developed model contains effective predictors to evaluate and manage the sleeping disorders including insomnia”.

     Moreover, we added a new subsection “6.3. Limitation, Application, and Future Research” (please see page 13 and lines 14-44) to discuss our results with future improvement as follows: “Despite the satisfactory results of the accuracy of the developed model, the presented study has several limitations. The first is the number and grouping of participants in the experiment. The experiment involved 22 males; all participants were considered conditionally healthy without sleep disorders and mental problems. In order to universalize the study and build models that are more difficult to classify, it is planned to expand the number of participants and include female participants and participants with sleep disorders such as insomnia or sleep anxiety in the experiment. Second, perceived quality of sleep was measured by PSQI questionnaire that was designed to define only two conditions - good and poor [55]. This parameter allowed us to build a binary model of sleep quality based on machine learning methods. In the future, it is planned to use several sleep questionnaires to expand the classification groups in mathematical modeling. Third, the presented research applied one type of objective physiological data – actigraph measures. This approach can be extended to use additional sensors to collect various types of physiological responses, for example, heart rate and galvanic skin response. The application of different sensory objective measures will be useful to improve the developed model. Also, in future research, it is planned to apply additional methods of deep learning, for example, a convolutional neural network. This will improve the analytical approach and find additional dependencies between objective and subjective measurements. The application of the obtained results and the developed model is possible in various theoretical and practical areas. Firstly, it expands knowledge in medical, behavioral, social and physiological sciences, including sensors of human physiological signals (actigraph, cardiograph and various stress indicators). New knowledge provides additional connections and dependencies between objective and subjective human data and assessments, and also allows for a better understanding of perceived well-being and health. The main area of the practical application of the results obtained is monitoring and developing additional approaches to diagnosing and tracking problems with the sleep process, as well as various mental, behavioral and physiological disorders. This is possible because many health problems are accompanied by poor sleep quality, such as anxiety, depression, eating disorders, as well as cancer and neurological diseases [44, 56]”.

  1. An external validation of your models in another population is suggested.

Answer) Thanks to the reviewer for the comment and suggestion. In the Table 5 and Section 6 “Discussion” (please see page 7) we previously explained and discussed the application and validation of MMASH dataset used in our paper for other studies and populations [30-32]. But, the authors agree that additional validation can be helpful for future studies. Based on this, additionally, we added a new subsection “6.3. Limitation, Application, and Future Research” (please see page 13 and lines 14-44). In this sub-section we described this population limitation (please also you can see this subsection citation in our above answer) and future research improvement.

  1. Sleep quality is also affected by the activities and life style in the daytime, these predictors are suggested to be included in the model.

Answer) Thanks to the reviewer for feedback and suggestions. The authors agree with the reviewer's opinion that daytime activity can affect sleep quality. However, in this study, we used the PSQI as ground truth for evaluating the proposed model’s performance. PSQI is a widely used ground truth for determining sleep quality in many studies. Our contributions are using the proposed models with limited data from actigraphy to predict the sleep qualities of subjects and connect objective (sensor) and subjective (PSQI) measurements in one model. New explanation on the connection between subjective and objective measurements we added as follows (please see page 11 and lines 25-36): “Previous research [36-43] proves the relationship between various sensory data and sleep parameters. However, in contrast to this literature, the present study links not only objective sensory measurements and sleep characteristics, but also subjective (personal) metrics of perceived sleep quality (PSQI survey results). These dependencies are important from the point of view of the health state and mental well-being of a person, since it is the perceived state of the sleep quality that determines the personal feeling of well-being [44]. Despite the fact that the data of such physiological sensors as an actigraph and a cardiograph provide sufficiently accurate data for assessing the sleep process, however, they reflect the physical state of a person to a greater extent. In turn, questionnaires (PSQI) allow us to evaluate the psychological component of the result of sleep (satisfaction and dissatisfaction) [44]. This is very important for identifying and diagnosing mental problems (anxiety, depression, and stress)”.

     The activities of daily living in the daytime can be considered and included in our future research.

  1. More of the article should be devoted to analyzing the strengths and weaknesses of the model used for the sleep quality analysis in this paper.

Answer) Thanks to the reviewer for feedback. According to this comment, the authors added more analysis on the strength and weaknesses of the developed model by including new correlation analysis, additional explanation and discussion. As the authors answered in previous comments (please you can see above), the authors included additional Pearson correlation analysis (please see page 8 and lines 13-16). Results of this new analysis were sowed in the above answers too; also, please see the page 9 and lines 35-38).

     The authors also showed more model reliability and discussed this by adding new text in sub-section “6.2. Model Performance” (please see page 12 and lines 36-53 or citation of this sub-section in the above answer).

     Also, as we mentioned, the authors added new subsection, “6.3. Limitation, Application, and Future Research” (please see page 13 and lines 14-44) to discuss our results (strength and weakness) with future improvement (please also you can see this subsection citation in our above answer).

 Minor concerns:

  1. The authors mentioned that “a homogeneous sample of subjects to minimize the differences between participants’ lifestyles and provide more objective results”, selection criteria of these participants and their characteristics should be presented.

Answer) Thanks to the reviewer for the comment. We used a previously proven experimental MMASH dataset with certain and previously selected participants by experimenters. Authors added participant characteristics as follows (please see page 5 and lines 24-26): “Twenty-two healthy young adult males (22 - 40 years old) were recruited in the experiment. Their weight and height are 60-115 Kg and 169-205 cm, respectively”.

  1. What is the population structure of the data source, and whether the model is suitable for all the population or only part of the population?

Answer) Thanks to the reviewer for feedback. 22 healthy young adult males (22 - 40 years old) were recruited. Their weight and height are 60-115 Kg and 169-205 cm, respectively. In the dataset, the female population is not included. Based on this, in future research, the proposed algorithm needs to be validated by female subjects and participants in more diverse population groups. The authors explained this in the newly added subsection “6.3. Limitation, Application, and Future Research” (please see page 13 and lines 14-44, also, you can see this subsection citation in our above answers).

  1. How does the actigraph determine and classify sleep and awaken status, whether there are errors, and whether errors have been well adjusted?

Answer) Thanks to the reviewer for the comment. Actigraph consists of accelerometer and inclinometer data recorded throughout the day. Basically, the actigraph provides the raw data of body movements so that the sleep-related diverse statuses cannot be provided by the actigraph itself. In order to determine and classify sleep, feature extraction and classification algorithms should be combined and integrated with actigraph.

  1. What is the limitation of the chose model, whether it can be further improved?

Answer) Thanks to the reviewer for feedback. According to the reviewer’s comment, the authors explained this in the newly added subsection “6.3. Limitation, Application, and Future Research” (please see page 13 and lines 14-44, also, you can see this subsection citation in our above answers).

Round 2

Reviewer 1 Report

I'm still not convinced of the benefits (or use cases) of predicting a binary (and subjective) measure of sleep. If the authors were predicting participants' subjective sleep scale, maybe they have a better case, but currently, I don't know how a binary (yes/no) sleep indicator is helpful in any context. The authors still do not provide an example of how this prediction is helpful. They cite papers that show how perceived sleep quality is related to other things, though I'd like the authors to provide a case where a clinician will make a decision based on the yes/no prediction provided by the model.

Author Response

Thanks to the reviewer for your comment. In the present study, we used the Pittsburgh Sleep Quality Index (PSQI), which was originally developed as a binary (good/poor quality) assessment tool. Based on the previous research, the authors found that even if the use of a binary classifier in some diagnoses is not enough, in any case, it can help significantly facilitate the task of the doctor. There are several studies and reports demonstrating the direct application of this binary assessment in medical practice. For example, “Test-retest reliability and validity of the Pittsburgh Sleep Quality Index in primary insomnia” (Backhaus et al., 2002) and “Association between poor sleep quality and depression symptoms among the elderly in nursing homes in Hunan province, China: a cross-sectional study” (Hu et al., 2020). These studies were conducted on real patient and elderly people data. In these studies, PSQI as a binary subjective assessment tool showed an ability to be a factor on the basis of which clinicians can conclude the presence/absence of insomnia or depression. It was shown that if PSQI reports poor sleep quality, it can correspond to health problems or disease presence. Based on studies, we can say that this binary subjective assessment of sleep quality can be considered separately as a factor in diagnosing sleep disorders or as a component for diagnosing mental problems, such as depression. These are two examples of how binary classification of perceived scores can be used in real medical practice. We have added this explanation in the “Discussion” section as follows (please see page 9 and lines 35-46): "Studies [45; 46] showed the medical application of binary subjective sleep quality assessment based on PSQI. Research [45] compared medical data of 80 patients with insomnia and a control group of 45 healthy participants. Authors reported that PSQI showed high reliability and good validity for patients with primary insomnia [45]. This result proves the ability of binary subjective sleep quality assessment to be used for sleep disorder detection. Mental health state was discovered based on PSQI in the study [46] using linear and binary logistic regressions. The authors collected PSQI and geriatric depression scale scores of elderly people in nursing homes to find the relations between subjective sleep quality and depression symptoms. The study supported the hypothesis that poor sleep quality is associated with increased depression signs in the elderly. Based on this, the authors presented the subjective sleep quality as one of the depression features".

Reviewer 3 Report

My concerns have been addressed. I have no further suggestions.

Author Response

Thanks to reviewer for kind feedback.